# Reframing Communication about Fall Prevention Programs to Increase Older Adults’ Intentions to Participate

**DOI:** 10.3390/ijerph21060704

**Published:** 2024-05-30

**Authors:** Meike C. van Scherpenseel, Lidia J. van Veenendaal, Saskia J. te Velde, Elise Volk, Di-Janne J. A. Barten, Cindy Veenhof, Marielle H. Emmelot-Vonk, Amber Ronteltap

**Affiliations:** 1Research Group Innovation of Human Movement Care, Research Center for Healthy and Sustainable Living, HU University of Applied Sciences Utrecht, 35011AA Utrecht, The Netherlands; saskia.tevelde@hu.nl (S.J.t.V.); di-janne.barten@hu.nl (D.-J.J.A.B.); cindy.veenhof@hu.nl (C.V.); amber.ronteltap@hu.nl (A.R.); 2Research Group Proactive Care for Older Adult People Living at Home, Research Center for Healthy an Sustainable Living, HU University of Applied Sciences Utrecht, 3501AA Utrecht, The Netherlands; lidia.vanveenendaal@hu.nl; 3Bachelor of Nursing, Institute for Nursing Studies, HU University of Applied Sciences Utrecht, 35011AA Utrecht, The Netherlands; 4Department of Rehabilitation, Physiotherapy Science and Sport, University Medical Center Utrecht, Utrecht University, 3508GA Utrecht, The Netherlands; 5Center for Physical Therapy Research and Innovation in Primary Care, Julius Health Care Centers, 3454 PV De Meern, The Netherlands; 6Department of Geriatrics, University Medical Center Utrecht, 3508 GA Utrecht, The Netherlands; m.h.emmelotvonk@umcutrecht.nl

**Keywords:** reframing, communication, fall prevention program, community-dwelling older adults

## Abstract

Introduction: Falls and fall-related injuries in community-dwelling older adults are a growing global health concern. Despite effective exercise-based fall prevention programs (FPPs), low enrollment rates persist due to negative connotations associated with falls and aging. This study aimed to investigate whether positive framing in communication leads to a higher intention to participate in an FPP among community-dwelling older adults. Methods: We conducted a two-sequence randomized crossover study. We designed two flyers, a standard flyer containing standard terminology regarding FPPs for older adults, and a reframed flyer highlighting fitness and activity by reframing ‘fall prevention’ as an ‘exercise program’ and ‘old’ as ‘over 65 years’. With a Mann–Whitney U test, we investigated group differences regarding the intention to participate between the flyers. A sensitivity analysis and subgroup analyses were performed. We conducted qualitative thematic analysis on open-ended answers to gain a deeper understanding of participants’ intention to participate. Results: In total, we included 133 participants. Findings indicated a significantly higher intention to participate in the reframed flyer (median = 4; interquartile range = 1–6) compared to the standard flyer (median = 2; interquartile range = 1–4) (*p* = 0.038). Participants favored more general terms such as ‘over 65 years’ over ‘older adults’. Older adults who were female, not at high fall risk, perceived themselves as not at fall risk, and maintained a positive attitude to aging showed greater receptivity to positively-framed communications in the reframed flyer. Additionally, already being engaged in physical activities and a lack of practical information about the FPP appeared to discourage participation intentions. Discussion: The results in favor of the reframed flyer provide practical insights for designing and implementing effective (mass-)media campaigns on both (inter)national and local levels, as well as for interacting with this population on an individual basis. Aging-related terminology in promotional materials hinders engagement, underscoring the need for more positive messaging and leaving out terms such as ‘older’. Tailored positively framed messages and involving diverse older adults in message development are essential for promoting participation in FPPs across various population subgroups to promote participation in FPPs among community-dwelling older adults.

## 1. Introduction

Falls and fall-related injuries among community-dwelling adults aged 65 years and older are increasingly common worldwide, making fall risk management and prevention an international health priority [1]. Fall-related injuries are associated with increased morbidity, physical limitations, and even mortality. Falls in older adults can also have serious socioeconomic consequences as they lead to increased health resource utilization [2]. Key elements of fall prevention management include routinely screening for fall risk factors, and subsequently referring to multidomain interventions that target a number of modifiable risk factors, such as muscle weakness, use of multiple medications, and home hazards [1,3,4]. A core element of multidomain fall prevention interventions is exercise-based fall prevention programs (FPPs), including balance, strength, and functional training [1]. Research indicates that such FPPs are effective in reducing both the rate and risk of falls among older adults living in the community [1,5]. Older adults who perceive themselves as at risk of falling, particularly due to higher age and a history of falls, experience recurrent falls and seem to be inclined to adhere to FPPs [2,6]. Nonetheless, the vast majority of older adults deny experiencing falls or being at risk of falling, resulting in a reluctance to participate in such programs and contributing to consistently low enrollment rates [2,7,8].

This lack of acknowledgment or denial of fall risk among many older adults are attributed to the negative connotations associated with falls. These negative associations encompass dependency, incompetence, functional impairment, psychological loss, and death [8,9]. Prior research into older adults’ perspectives on falling has revealed that many older adults reject the label of being ‘at risk of falling’, due to these negative connotations [8,9,10]. Moreover, they perceive information and fall prevention interventions, such as FPPs, as applicable to others, but not relevant to themselves personally [9,10,11,12]. Acknowledging falls as personally relevant would threaten their identity as being autonomous, competent, and independent [9,10,11]. Especially those with good mental and physical health and no or a limited history of falls, are most likely to reject their personal fall risk. ‘Falling’ among older adults can thus be considered a stigmatized topic [13,14]. This social stigma has been recognized to have a significant impact on older adults’ willingness to report and discuss falls, and to enroll in an FPP [2,9,13,15].

Besides not acknowledging being a ‘faller’, older adults also tend to actively dissociate themselves from being labeled as ‘old’, since being old has been associated with similar negative traits as those related to falls [16]. The prevailing social stereotype of aging is negative, especially in Western society [17]. Individual attitudes to own aging can have effects on health behaviors or outcomes [18]. Negative attitudes to aging, i.e., when the process of aging is viewed unfavorably, have been shown to have negative effects on the health and well-being of older adults [19,20]. Positive attitudes, on the other hand, are associated with higher life satisfaction and better physical and mental health [18,21]. Successfully addressing the stigma around ‘falls’ and being ‘old’ and refocusing on these topics from a more positive perspective may be a valuable strategy to promote participation in FPPs among older adults [14]. Research has suggested that the use of terminology such as ‘fall prevention’ and ‘falls’ is problematic, since these terms are negatively associated with aging [8,9]. Communication incorporating terms related to fall prevention should be replaced by more positive messages, such as emphasizing health and independence, to create awareness, and increase engagement in fall prevention interventions [8,9,22]. In general, positive communication about health is seen as a key strategy to inform, motivate, and eventually achieve optimal health behaviors [23,24,25]. In addition, previous research has shown that positive messaging about being ‘old’ reduces implicit negative attitudes, and older adults with more positive attitudes toward aging tend to practice more preventive health behaviors [26,27]. Furthermore, well-designed communication visuals of health messages can increase attention and adherence to the proposed health instructions [28,29].

Hence, research has shown that positively framed messages can enhance older adults’ awareness and adoption of fall prevention interventions, and generally, positive messaging promotes health behaviors. However, it remains unclear whether positively reframing ‘fall prevention’ and ‘aging’ may increase participation rates in FPPs among community-dwelling older adults. In addition, it is not known whether and how demographic factors and perceptions might moderate the potential effectiveness of positively framed communication on an intention to participate in FPPs. Therefore, in the current study, our primary objective was to investigate whether positive framing in communication leads to a higher intention to participate in an FPP among community-dwelling older adults. Secondly, we hypothesized that the potential effectiveness of positive communication in influencing the intention to participate would be moderated by various factors, such as age and attitudes to aging.

## 2. Materials and Methods

### 2.1. Study Design

A two-sequence randomized crossover study was conducted to achieve both research objectives (Figure 1). The current study is part of the larger, multi-disciplinary Dutch implementation study on the implementation of fall prevention in the community, FRIEND (fall prevention implementation study). Both FRIEND and this sub-study received ethical clearance from the Ethical Committee Research Healthcare Domain of the HU University of Applied Sciences, Utrecht, the Netherlands (reference number: 113-001-2020 and 225-000-2023, respectively). This study is reported following the strengthening the reporting of observational studies in epidemiology (STROBE) guidelines [30].

### 2.2. Participants and Recruitment

Participants were eligible if they: (1) were aged 65 years or older; (2) were living at home; and (3) were able to read and understand the Dutch language. Participants were recruited between March and September 2023, in three ways. First, a link to a digital form was distributed via social media. Older adults who were interested in participating in the study were asked to only fill out the digital form if they fulfilled the inclusion criteria. Potential participants could leave their names and home addresses, after which they received an envelope, containing an information letter, informed consent forms, and a questionnaire in their mailbox at home. In the information letter, we called our study the “FIT-study” to prevent participants from searching for the FRIEND study and realizing they were involved in a study about fall prevention, which could potentially bias the response rate. Second, researchers of the FRIEND research group handed out envelopes during social/physical activities for community-dwelling older adults in neighborhoods in the area of Utrecht, the Netherlands. Third, a physiotherapist, actively participating in the FRIEND project, distributed envelopes at her primary care practice. As an extra incentive for participation, five generally valid vouchers worth €20 were raffled among all participants. All participants provided their written informed consent to participate in this study.

### 2.3. Study Procedure

In this study, we investigated whether one particular strategy (positively reframing ‘fall prevention’ and ‘aging’) can effectively promote participation in an FPP among community-dwelling older adults. This strategy was designed based on prior stages of the FRIEND project [14].

In the current study, we designed two flyers based on the abovementioned strategy, differing only in their textual content. This allowed us to specifically investigate the result of using different terminology to promote participation. The standard flyer (titled “Prevent Falling”) contained standard terminology such as ‘fall prevention programs’ for ‘older adults’, i.e., terms perceived as stigmatizing according to our previous research. In contrast, in the reframed flyer (titled “Keep Moving”) we positively reframed these concepts, highlighting fitness and activity by reframing ‘fall prevention’ as an ‘exercise program’ and ‘old’ as ‘over 65 years’ (Appendix A).

Prior to data collection, we pre-tested the study materials among community-dwelling older adults who met the inclusion criteria (n = 8). The objective was to ensure that the flyers were indeed differently framed. Using a 5-point Likert scale (1 = not at all, 5 = extremely) we posed four questions to evaluate whether the topics ‘fall prevention’ and ‘aging’ were emphasized in the standard flyer and were absent in the reframed flyer (Appendix A). Next, we compared the mean scores of all questions and set the mean cut-off score at 3 points (neither agree nor disagree). Analysis revealed that the standard flyer consistently scored ≥ 3 points for all questions and the reframed flyer scored ≤ 2.75 points. This indicated that the reframing was successful.

After enrollment in the study, participants were randomized into either the AB or the BA sequence (Figure 1). Randomization was performed by hand. The envelopes were distributed alternately. Participants were instructed to fill out the questionnaire alone at home. The questionnaire consisted of three parts (Table 1). Participants were asked to first fill out Part I, which contained questions on background characteristics, self-perceived general health [31], self-perceived risk of falling [6], fall risk indicators (fall in past 12 months; number of falls in the past 12 months; difficulties with moving, walking, and balance) [32], and participated in an FPP before and the Attitudes To Aging Questionnaire-Short Form (AAQ-SF) [33]. Based on the fall risk indicators, we were able to determine whether participants were at high risk of falling, having experienced two or more falls in the past 12 months and/or having difficulties with moving, walking, and balance [32]. The AAQ-SF is a 12-item self-reported questionnaire consisting of three subscales: physical change, psychosocial loss, and psychological growth, each with four questions [33]. The items of the AAQ-SF are scored on a 5-point Likert scale, resulting in each subscale having minimum scores of 4 and maximum scores of 20 [18,33]. The English questionnaires were all translated into Dutch using the forward–backward translation method [34].

After filling out Part I, participants were asked to open a small envelope that contained both the first flyer of their sequence and Part II (Table 1). Part II contained the 7-point Likert scale, the Attitude Toward the Ad (Affective) Scale, with 3 subscales [35], and a 7-point Likert scale question on the intention to participate in the program within the next 6 months (‘Do you have the intention to participate in this program within the next 6 months?’). Participants were asked to clarify their answers in an open field. Participants were then asked to put the flyer and Part II back in the small envelope and seal it. Sealing the envelope before looking at the second flyer ensured that participants could not compare flyers and/or change their initial answers. Then, they were instructed to open the second small envelope, which contained the other flyer and Part III (identical to Part II). After filling out Part III, participants were again asked to put the flyer and Part III back and seal the envelope. A return envelope with a preprinted university address was then used to return the questionnaire. All data were manually inserted into a digital format of the questionnaire, of which a few were inserted by two researchers to check the correctness of data entering.

### 2.4. Statistical Analysis

Before the start of recruitment, we calculated the sample size needed to compare two paired proportions, in line with the two-sequence crossover design, using a G*Power tool [36]. A previous study showed that a proportion of 16% of older adults intended to participate in an FPP [9]. We therefore hypothesized that, in total, 16% of the older adults would indicate a positive intention to participate after they were presented with the standard flyer, while in total 25% of the older adults would have a positive intention to participate after being presented with the reframed flyer. Next, we assumed that of the proportion of 16% intending to participate after being presented with the standard flyer, the majority (75%) would also indicate positive intention after being presented with the reframed flyer. The proportion of discordant pairs resulted in 0.3. With an α of 0.05 and a power (1-β) of 0.8, and with a two-tailed sample size estimation, we aimed to recruit 210 participants in this study to show an effect.

Statistical analyses were performed using IBM SPSS Statistics^®^, version 29.0.0.0. All items from Part I (background characteristics) were analyzed using descriptive statistics. The Likert scale and categorical data were summarized descriptively using frequencies (percentages) and continuous data were described using means and standard deviations. We identified participants having a high fall risk based on the fall risk indicators, i.e., experiencing more than one fall in the past 12 months and/or having difficulty with mobility, walking, or balance [32]. Furthermore, attitude to aging was assessed by analyzing the subscales of the AAQ-SF to ascertain whether participants leaned towards positivity or negativity. The original 24-item AAQ, consisting of 8 items per subscale, established cut-off scores to define the positive range per subscale, with a threshold of 24 points suggesting an individual item score of 3 (neither disagree nor agree) [18]. The subscale psychosocial loss was negatively worded, so cut-off scores for a positive attitude to aging on the individual subscales were defined as physical change ≥ 24 points; psychosocial loss ≤ 24 points, and psychological growth ≥ 24 points. As the AAQ-SF is half the size of the original scale, and no cut-off values have been specified in the literature, we adopted a cut-off score of 12 points in this study. Subscale scores were derived by adding scores of all four items. Next, the predefined cut-off score was used to categorize participants into positive or negative ranges for each subscale, i.e., aspects of attitude to aging.

To analyze whether the intention to participate within the next six months differed between the exposure to the standard flyer and the reframed flyer, we originally intended to employ a paired-sample McNemar’s test. However, considering potential carry-over effects [37], which could have resulted in biased responses to the second flyer after having seen the first flyer, we opted to evaluate our main objective by comparing participants’ responses after receiving their initial flyer. Specifically, we compared participants who received the standard flyer first (sequence AB) with those who received the reframed flyer first (sequence BA). To do so, we conducted a non-parametric Mann–Whitney U test (between-subjects) [38]. This test was suitable considering the non-normal distribution of the data, which was confirmed through tests of normality, the Kolmogorov–Smirnov and Shapiro-Wilk tests. Missing data were examined; small fractions (≤10%) of missing data of the outcome variable resulted in cases with missing values being excluded from analysis (complete case analysis) [39]. Next, we conducted a sensitivity analysis to ascertain whether the flyer with a higher intention to participate consistently scored higher, regardless of its presentation order in the AB or BA sequence. This was achieved by employing a non-parametric Wilcoxon signed rank test for participants in the AB and BA sequences. We compared the median scores between the standard and reframed flyer within-subjects for individuals in these sequences. In addition, the mean difference scores of all participants were calculated to gain insight into the mean differences of the scores on ‘intention to participate’ for both flyers in the same participants. Furthermore, descriptive analyses were conducted to explore differences between the standard flyer and the reframed flyer regarding the score of the Attitude Toward the Ad (Affective) Scale. Given the change of tests for our primary aim, we conducted a post hoc power analysis using G*Power software to estimate the power of the used tests [36].

Secondary analyses were performed by multiple independent subgroup analyses. This allowed us to investigate whether potential moderating factors influenced the effectiveness of positive communication on the intention to participate. This allowed us to gain a deeper understanding of how the effect of the flyers might vary across different subgroups. First, the data were split into subgroups based on the moderator variables of interest: age group, gender, self-perceived risk of falling, history of falling, high fall risk, self-perceived general health, previous participation in an FPP, and attitude to aging [2,6,21,40,41]. For the comparison of subgroups, it is necessary to define categories in continuous data. Therefore, age was divided into three life-stage subgroups: the young-old (65–74 years of age), the middle-old (75–84 y), and the old-old (over 85 y) [42,43]. Separate Mann–Whitney U tests were conducted per subgroup to investigate differences in the outcome variable ‘intention to participate’ between the standard and reframed flyer. Next, descriptive statistics (median, interquartile range (IQR)) of the intention to participate were computed within the subgroups for both the standard and reframed flyer to further explore the differences between the flyers. The significance of all tests was defined at the level of *p* ≤ 0.05. Multiple testing correction was performed to avoid Type I errors by controlling the expected proportion of falsely rejected hypotheses, the false discovery rate (FDR), using the Benjamini–Hochberg (BH) procedure. In this procedure, the original *p*-value is compared to the adjusted critical value calculated by the BH procedure. If the original *p*-value is smaller than the BH-adjusted *p*-value, it indicates that the result remains significant after correcting for multiple tests. The FDR threshold was set at 0.1 [44,45].

Lastly, the open-ended responses provided by the participants following their score on the intention to participate of each flyer were thematically inductively analyzed by one researcher (MS) to gain a deeper understanding of the answers related to the outcome variable ‘intention to participate’. Initially, open codes were generated, followed by axial coding to establish subthemes and subsequently overarching themes [46].

## 3. Results

In total, 236 participants received an envelope, of which 153 envelopes were returned (64.8%). Recruitment via social media yielded the largest number of responses (n = 109). Due to missing signed informed consent forms, 20 questionnaires were excluded from further analyses. This resulted in a total of 133 participants whose questionnaire data were used in the analyses. Of these 133 participants, 4 questionnaires (3.1%) were not completed regarding the outcome variable ‘intention to participate’, leading to 129 complete cases, of which 62 and 67 participants were included in the AB and BA sequence, respectively.

The background characteristics of the study population are presented in Table 2. The majority of the participants were female (66.2%), were living together (69.2%), were higher educated (58.6%), and scored their own health as moderate to good (94.6%). Of all participants, 26.3% experienced a fall in the past 12 months, whereas 15.3% perceived they were at risk of falling. In total, 11.4% had participated in a FPP before. The majority of participants scored within the positive range of attitudes to aging across all three subscales (94.4%, 96.0%, and 95.2%, respectively).

### 3.1. Primary Analysis: Differences in Intention to Participate within the Next Six Months between the Standard Flyer and the Reframed Flyer

A non-parametric Mann–Whitney U test (between-subjects) was conducted. Findings revealed significantly higher scores on the intention to participate for the reframed flyer (median = 4; IQR = 1–6) compared to the standard flyer (median = 2; IQR = 1–4) (*p* = 0.038).

Next, sensitivity analysis (within-subjects) was performed using a non-parametric Wilcoxon signed rank test. Analysis indicated that the differences in scores for the intention to participate did not display statistical significance within-subjects between the flyers for both the AB and BA sequences (*p* = 0.793 and *p* = 0.273, respectively). Mean difference scores between both flyers within-subjects were computed for all participants, showing that the majority of the participants in the same sequence scored both flyers similarly (mean = 0.08, standard deviation = 1.93). This indicates that the reframed flyer scores higher on the intention to participate only when presented separately from the standard flyer.

Descriptive analyses of the Attitude Toward the Ad (Affective) Scale revealed that the reframed flyer consistently received higher scores across all subscales, in comparison to the standard flyer (Table 3). The largest disparity was observed in the final subscale (doesn’t excite me/excites me), whereas variances in the first two subscales were minimal. Post hoc power calculations were performed, resulting in a power of 0.78.

### 3.2. Secondary Analyses: Moderating Factors of the Effectiveness of Positive Communication on Influencing Intention to Participate

After splitting the data into subgroups, analyses revealed several differences between subgroups (Table 4). We found statistically significant differences in the intention to participate between the standard flyer and the reframed flyer, in favor of the reframed flyer, among female participants (*p* = −0.020), among participants with no perceived fall risk (*p* = 0.039), among participants who objectively, i.e., based on guidelines, did not have a high fall risk (*p* = 0.010), and among participants who scored in the positive range of the ATA-SF subscales (*p* = 0.021, *p* = 0.045, and *p* = 0.014 for subscale physical change, psychological growth and psychosocial loss, respectively). After adjusting the *p*-values using the Benjamini–Hochberg correction at the specified FDR threshold, no original *p*-values remained significant.

### 3.3. Qualitative Inductive Thematic Analysis

The majority of the participants provided additional reasoning for their scoring rate on the intention to participate for the standard flyer (“Prevent Falling”) and reframed flyer (“Keep Moving”).

#### 3.3.1. Theme 1: This Program Does Not Apply to Me

This theme was present among nearly all participants who scored low on the intention to participate (score 1–3) in the standard flyer. These individuals frequently attributed their reluctance to participate in an FPP to a lack of the presence of health issues or perceived problems with falling. Moreover, there was a perception that the program catered primarily to the ‘oldest old’, typically those aged over 80, rather than to themselves. In addition, some participants mentioned their already active lifestyles and current engagement in physical activities, such as walking, biking, or swimming, as reasons for their reluctance to engage.

Being physically active was highlighted by participants across all scores in response to the reframed flyer as a reason for not intending to participate in the promoted program. In addition, some participants acknowledged a few positive aspects of the reframed flyer, noting that it targeted a broader group beyond just ‘older adults’ and emphasized a broader perspective beyond fall prevention alone.

#### 3.3.2. Theme 2: Being Physically Active Is Important

Individuals scoring moderate to high on the intention to participate (4–7 points) in the standard flyer underlined the significance of addressing the issue of fall prevention. Some expressed uncertainty about their mobility capabilities or feared major consequences in case of a fall. Some emphasized the necessity of participating in an FPP, viewing it as crucial not only for people aged over 65. Conversely, those scoring moderate to high on the reframed flyer highlighted the importance of exercising to maintain overall fitness and health in older age.

#### 3.3.3. Theme 3: Organizational Aspects of the Program

Some participants mentioned that participation in FPPs depends on the organizational aspects of the program. For instance, they emphasized the program has to be conveniently located within their neighborhood and scheduled during midday, rather than in the evenings. In addition, preferences varied among participants, with some preferring group programs, while others favored a more individual approach. Suggestions were made to integrate elements of fall prevention exercises into physical activity programs in which older adults already participate, to lower the threshold to participate.

#### 3.3.4. Theme 4: Design of the Flyer

Overall, some participants noted that neither flyers encouraged their participation in either program. Regarding the standard flyer, participants mentioned that it did not appeal to them since the text had a patronizing tone and negative wording, which failed to generate interest. For both flyers participants highlighted that the text was too lengthy but lacked important information such as location, date, costs, group/individual, and trainer contact details. This lack of clarity about the program was one of the reasons for being unwilling to participate.

Moreover, some participants found the visual aspect of the flyer uninspiring. They described it as dull and unremarkable, with an unattractive, unappealing illustration and pale colors.

Suggestions were made to improve the design of the flyers, such as emphasizing more extensively that it is a fun group program and that it adds pleasure and joy to physical activity by exercising together with peers.

## 4. Discussion

The primary aim of this study was to explore whether positively framed communication leads to higher intention to participate in an FPP within the next six months among community-dwelling older adults. Our secondary aim was to examine whether the effectiveness of positive communication on the intention to participate would be moderated by various factors, such as age and attitudes to aging.

Findings on our primary aim indicated that older adults are more inclined to participate in an exercise program, such as an FPP, when they are provided with positively framed communication concerning ‘fall prevention’ and ‘aging’, compared to standard communication typically used to promote participation in such programs. The reframed flyer almost always scored higher on the intention to participate as opposed to the standard flyer across various subgroups. Our findings are in alignment with previous research, which also highlighted the efficacy of positively framing messages in promoting general physical activity behavior among older adults, resulting in increased participation in physical activities [24,47,48]. Our results also fit the so-called ‘positivity effect’, which appears to be prevalent among older adults [49]. Older adults show a preference for positive over negative information, and they forget negative information more quickly than positive information across a wide range of materials, such as images and health-related messages [25,49,50]. Similarly, regarding fall prevention, older adults seem to prefer positive communication that focuses on health, independence, and the benefits of training, rather than the risks of falling [9,11,22].

Our secondary aim’s outcomes showed an increase in the intention to participate among specific subgroups of individuals, when presented with the reframed flyer. Although the results did not remain significant after correction for multiple tests, they still present potentially relevant findings. Notably, females seemed to have a higher intention to participate with the reframed flyer compared to the standard flyer. This aligns with prior research indicating gender influences older adults’ attitudes toward fall prevention and their willingness to engage in fall preventive measures, such as FPPs [51]. Studies indicate older women are more likely to attend fall prevention group programs, and are more receptive to fall prevention messages than men [52]. Conversely, older men seem to perceive exercise programs as feminine, suggesting program advertisements featuring men would increase participation rates [53]. Unfortunately, research on how message differences impact adherence rates in FPPs is limited, emphasizing the need for further investigation to inform gender-specific marketing strategies for FPPs [52].

Findings suggest that the reframed flyer could be more appealing to self-perceived active, healthy, and optimistic older adults. In particular, the reframed flyer received higher scores for the intention to participate from individuals with positive attitudes to aging, and those with no high fall risk or self-perceived fall risk, although results were not statistically significant. Qualitative findings showed that some of these individuals favored the reframed flyer for its broader perspective on exercising and fitness. In addition, they found the standard flyer uninteresting as it solely emphasized fall prevention, contradicting their perceived absence of fall risk. These perceptions are in line with previous research, emphasizing that the negative association of falls leads to older adults denying and underestimating their personal fall risk, resulting in a rejection of participating in FPPs as they consider it as not being necessary for them [8]. Moreover, individuals with good or moderate self-perceived general health scored the reframed flyer higher, aligning with previous research on the positive link between self-reported beneficial health and participation in exercise programs among healthy older adults [40]. It is essential to highlight the importance of fall prevention exercise participation specifically, rather than engaging in regular physical activity. International guidelines emphasize that physical activity alone is not enough to prevent falls, with exercise programs incorporating balance, functional exercises, and strength training being particularly effective in reducing fall risk [1,54]. Nevertheless, in our study, physically active (walking, biking, swimming) adults expressed reluctance to enroll in the promoted programs, perceiving other activities as sufficient for health benefits as participation in FPPs [12]. Integrating fall prevention-specific exercises into existing physical activities could benefit physically active, optimistic older adults, potentially reducing their fall risk without drawing attention to it.

In contrast, the standard flyer, which specifically mentioned ‘fall prevention’, seemed to be more persuasive to participate in an FPP to individuals who perceive themselves at risk of falling, have poor self-perceived general health, are at high risk of falling, and hold negative attitudes to aging. These older adults emphasized the importance of addressing the issue of fall prevention. Although not statistically significant the findings could indicate that those perceiving themselves as fallers, having health issues and negative attitudes to aging, are more receptive to communication targeting ‘fall prevention’.

These insights imply that it would be beneficial for professionals to identify specific demographics and characteristics of the population they are targeting, allowing them to tailor communication accordingly to individuals with a range of different abilities, ages, and lifestyles. Prior research has also shown positive effects of message tailoring to encourage health behavior change, such as engaging in physical activity [24,55,56].

In our study, flyers were utilized to promote participation in an FPP. Prior research showed mixed responses to flyers, with some preferring direct information from healthcare professionals while others believed flyers could potentially initiate discussions about fall prevention [14]. Print brochures and flyers could be used as supplement materials to healthcare providers’ information, and may be effective in capturing attention [2,22,57]. Additionally, well-designed visual materials have been found to enhance communication, attention, and adherence to health instructions [29]. In the current study, the reframed flyer received higher scores in terms of attractiveness, although both flyers were generally lacking in visual appeal. Despite this, differences were still evident, suggesting that well-designed materials might increase between-group differences. Moreover, to effectively design (audio-)visual material aimed at enhancing fall prevention awareness and promoting engagement among the broader older population, it is recommended to involve a diverse group of older adults in message development [58].

However, it should be noted that positively reframing communication alone using a flyer may not be sufficient to promote participation among all community-dwelling older adults. An 18-month multi-media campaign in Australia aimed at reducing falls among older adults by promoting physical activity revealed that only 11% became aware through a pamphlet. In addition, only 22% actually became more active as a result of the campaign [59]. Organizational factors such as program duration, location, and format (group or individual) may also influence engagement in FPPs, as highlighted in the current and prior research [14]. Future research could explore combining strategies in promoting participation in FPPs among community-dwelling older adults, providing more comprehensive approaches for older adults. For example, combining communication strategies with program tailoring, providing technical assistance, and consciousness-raising strategies [60].

It is noteworthy that the large majority of participants in our study scored positively across all subscales of the Attitudes to Aging Questionnaire-Short Form, indicating a generally positive perception of the aging process. This trend has been seen in prior studies as well, suggesting that the individual experience of aging is often seen in a more positive light [17,18,19]. Positive attitudes to aging among older adults have been associated with diverse factors, such as higher education, female gender, and cohabitation [61,62]. In our study, most participants were highly educated (58.6%), were living together (69.2%), were female (66.2%), were relatively young (mean = 74.4 years), and reported moderate to good self-perceived health (94.6%), which may explain the large proportion of older adults reporting positive attitude to aging observed in our sample.

Moreover, perceptions of aging among older individuals influence the decision to participate in physical activities and exercise [18,20,39]. Those with more positive attitudes tend to have better physical and mental health outcomes, and practice more preventive health behaviors such as taking exercises [18,26,41]. It has been hypothesized that this can be explained by their stronger beliefs in the need to take care of their own health [18]. Our findings reflect this trend, with most participants expressing positivity about aging while also engaging in physical activities to maintain fitness. Future research should investigate how older adults arrive at these positive perceptions, in order to develop policies to promote more positive attitudes to aging [18].

Interestingly, despite a high proportion of participants displaying positive attitudes to aging in our study, age-related stigma endured. This is illustrated by many participants’ perception that the standard flyer’s description of the FPP for older adults was targeting only the ‘oldest old’, leading the majority to feel it was unsuitable for them. This suggests a difference between stigma and attitudes to aging, as discussed in prior literature [63,64]. While self-perceived stigma involves an age-based stereotype threat that older adults perceive from external social groups, attitudes to aging reflect older adults’ internal views on aging, including subjective beliefs about their cognitive and physical capabilities [63]. Despite the external stigma, older adults may maintain a positive attitude toward their own aging, serving as a buffer against negative stereotyping in society [19,63]. In our study, participants preferred the reframed flyer’s terminology used to describe aging, favoring more general terms such as ‘over ‘65 years’ over ‘older adults’. This suggests that even those with positive attitudes to aging may not feel encouraged to participate in FPPs when stigmatizing terminology related to ‘old’ is used. Therefore, it is recommended to avoid age-related terminology that carries negative connotations when promoting broad participation in FPPs (i.e., including healthier older adults).

There are some strengths to this study. Firstly, to our best knowledge, this is the first study to explore the impact of reframing stigmatizing topics about fall prevention and aging on the intention to participate in FPPs, addressing a gap in the literature. Given the increasing attention to fall prevention management in aging populations, the findings are particularly meaningful for (inter)national mass-media campaigns, local initiatives, and professionals in their interactions with older adults. Our study offers practical implications for policy and practice. Secondly, secondary analyses and thematic analysis provided rich insights into results among various subgroups, resulting in a deeper understanding and better interpretation of the findings. Thirdly, the utilization of validated questionnaires ensured the accuracy and consistency of data collection, enhancing the credibility of study outcomes and comparability with other studies. Lastly, the randomization of participants minimized confounding variables, enhancing internal validity and the reliability of the findings.

There are also limitations to consider. Firstly, selection bias may have influenced the study, resulting in a predominantly highly educated study population with moderate to good self-perceived health and positive attitudes to aging. Recruitment primarily through social media platforms could have contributed to this bias, which is commonly utilized by younger or middle-aged older adults, the highly educated, and women [65]. A broader distribution of the questionnaire through non-digital channels might have diversified the characteristics of the study sample. This could have facilitated more robust secondary analyses, as currently, some subgroups lacked sufficient sample sizes for meaningful results. Additionally, the study’s title, “FIT-study”, and description may have attracted individuals predisposed to the topic, potentially further biasing the sample. Moreover, it is more common for highly educated and younger individuals to participate in research projects, further contributing to this bias [66]. Therefore, findings have to be carefully interpreted as the sample may not fully represent the population of interest. Future research could explore the effectiveness of positively framed communication among a more diverse sample, including older adults aged 80+, with lower educational levels, poorer self-perceived health, and more negative attitudes toward aging. These vulnerable populations are particularly important to engage in FPPs due to their higher risk of falls, yet existing evidence primarily focuses on populations not representative of them [67,68].

Secondly, we employed a two-sequence randomized crossover design with the assumption we could compare paired proportions of positive intentions. However, we encountered potential carryover effects, indicating that the scores of the first flyer influenced the scores of the second due to the absence of a washout period [37]. Thus, we opted for the comparison of two independent samples using the scores on the first flyer only. Despite the fact that we did not reach the original sample size needed, post hoc power calculations showed a power of 0.78, which closely approached the desired power of 0.8.

## 5. Conclusions

In conclusion, our study highlights the importance of positively framed communication regarding fall prevention and aging to promote participation in FPPs. Negative associations with aging-related terminology may hinder engagement, underscoring the need for more positive messaging. While messages specifically mentioning ‘fall prevention’ may not resonate with positive, active older adults, it does appeal to those who perceive themselves to be at risk of falling. Additionally, given that engagement in regular physical activities may discourage participation intentions, integrating specific fall prevention exercises into existing activities could stimulate participation.

These findings provide practical insights for designing (mass-)media campaigns at both (inter)national and local levels, as well as for individual interactions with this population. Our results emphasize the importance of creating well-designed, tailored positively framed communication materials to promote participation in exercise-based FPPs, while avoiding negative and stigmatizing terminology.

## Figures and Tables

**Figure 1 ijerph-21-00704-f001:**
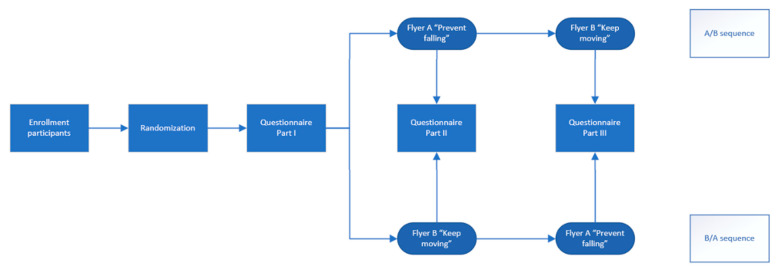
Two-sequence randomized crossover design in the current study.

**Table 1 ijerph-21-00704-t001:** Content of questionnaires parts I–III.

Questionnaire	
Part I	- Background characteristics: Age Gender Household composition Educational level - Self-perceived general health (1 = very bad, 4 = good) - Self-perceived risk of falling (yes/no) - Fall risk indicators: Fall in past 12 months (yes/no) Number of falls in the past 12 months Difficulties moving, walking, balance (yes/no) - Participated in an FPP before (yes/no) - AAQ-SF (1 = strongly disagree, 5 = strongly agree) Physical change Psychological growth Psychosocial loss
Part II, Part III	- Attitude Toward the Ad (Affective) Scale (1 = strongly disagree, 7 = strongly agree) Catchy/not catchy Appeals to me/doesn’t appeal to me Excites me/doesn’t excite me - Intention to participate in the next six months (1 = strongly disagree, 7 = strongly agree)

Abbreviations: FPP = fall prevention program; AAQ-SF = Attitudes to Aging Questionnaire-Short Form.

**Table 2 ijerph-21-00704-t002:** Background characteristics of the participants in the study.

Background Characteristic	Total Sample (n = 133)
Age, mean (SD)	74.4 (± 5.8)
Gender, female (%)	66.2%
Household composition (%) - Alone - Together with partner - Together with partner and child(ren) - Together with child(ren)	30.8% 64.7% 1.5% 3.0%
Educational level (%) - Primary school - Secondary school - Secondary vocational education - Higher professional education - University	4.5% 18.8% 18.0% 46.6% 12.0%
Self-perceived risk of falling, yes (%)	15.3%
Self-perceived general health, (%) - Good - Moderate - Poor - Very poor	35.1% 59.5% 4.6% 0.8%
Fall history past 12 months, yes (%)	26.3%
If yes, how many times did you fall, mean (SD)	1.7 (±1.05)
Difficulty with mobility, walking or balance, yes (%)	22.0%
High risk of falling *, yes (%)	29.3%
Participated in FPP before, yes (%)	11.4%
Attitudes To Aging Short Form - Physical change, positive (%) - Psychological growth, positive (%) - Psychosocial loss, positive (%)	94.4% 96.0% 95.2%

Abbreviations: SD = standard deviation; FPP = fall prevention program. * According to Dutch Guidelines: based on experienced fall past 12 months > 1 and/or difficulty with mobility, walking, or balance = yes.

**Table 3 ijerph-21-00704-t003:** Results from descriptive analysis of the Attitude Toward the Ad (Affective) Scale.

	Standard Flyer	Reframed Flyer
Subscale Attitude Toward the Ad (Affective) Scale (range 1–7), mean (SD) - Not catchy/catchy - Doesn’t appeal to me/appeals to me - Doesn’t excite me/excites me	4.15 (±1.740) 4.75 (±1.691) 3.85 (±1.691)	4.31 (±1.569) 4.78 (±1.525) 4.12 (±1.472)

Abbreviations: SD = standard deviation.

**Table 4 ijerph-21-00704-t004:** Results of the secondary analyses, including multiple testing corrections using the Benjamini–Hochberg procedure.

	Standard Flyer (A), Score Intention to Participate in the Next Six Months (Median, IQR; n *)	Reframed Flyer (B), Score Intention to Participate in the Next Six Months (Median, IQR; n)	*p*-Value	BH Adjusted *p*-Values
Age - 65–74 years of age - 75–84 years of age - 85+ years of age	2 (1–4; n = 34) 2 (1–6, n = 26) 2 (1–2, n = 2)	3.5 (1–4.75; n = 36) 5 (1–6.75; n = 24) 2 (1–7; n = 7)	0.074 0.179 0.877	0.041 0.064 0.100
Gender - Female - Male	2 (1–4; n = 41) 2 (1–5; n = 21)	4 (1.25–6; n = 44) 2 (1–5; n = 23)	**0.020** 0.855	0.014 0.095
Self-perceived risk of falling - Yes - No	4 (2–6; n = 11) 2 (1–3.75; n = 48)	5 (2.5–6.75; n = 8) 3.5 (1–6; n = 54)	0.498 **0.039**	0.082 0.023
Self-perceived general health - Good - Moderate - Poor - Very poor	1 (1–3; n = 25) 2 (1–4.75; n = 32) 4.5 (2.25–6.75; n = 4) 2 (2–2; n = 1)	4 (1–6; n = 21) 4 (1–6; n = 42) 2 (2–2; n = 2) n/a	0.066 0.150 0.140	0.036 0.059 0.050
Fall history in the past 12 months - Yes - No	2 (1–3.75; n = 16) 2 (1–5; n = 46)	4 (1–6; n = 19) 4 (1–6; n = 48)	0.181 0.099	0.068 0.045
High risk of falling - Yes - No	3 (1.5–6; n = 17) 2 (1–3.5;n = 45)	4 (1–6; n = 21) 4 (1–6; n = 46)	0.833 **0.010**	0.091 0.005
Participated in an FPP before - Yes - No	1 (1–5.5; n = 5) 2 (1–4; n = 57)	4 (1–6.25; n = 10) 4 (1–6; n = 56)	0.408 0.055	0.077 0.032
Attitudes to aging—Short Form - Physical change ○ Positive range ○ Negative range - Psychological growth ○ Positive range ○ Negative range - Psychosocial loss ○ Positive range ○ Negative range	2 (1–4; n = 57) 4.5 (2.25–6.75; n = 4) 2 (1–4; n = 59) 1 (1–1; n = 3) 2 (1–4; n = 58) 4.5 (1.50–6.75; n = 4)	4 (1–6; n = 58) 2 (1–2; n = 3) 4 (1–6; n = 59) 4 (1–4; n = 2) 4 (1–6; n = 59) 1 (1–1; n = 2)	**0.021** 0.212 **0.045** 0.739 0.014 **0.140**	0.018 0.073 0.027 0.086 0.009 0.055

Abbreviations: BH = Benjamini–Hochberg; IQR = interquartile range; n/a = not applicable. * n differs between subgroups due to missing data. Bold = statistically significant (α ≤ 0.05).

## Data Availability

The raw dataset generated and utilized in this study is archived through the repository DataverseNL (https://doi.org/10.34894/EWR6C0) and is available upon request by contacting the corresponding author. The general description and syntax of the generated data and informed consent forms are publicly available. Note that the data provided are anonymous to ensure privacy and confidentiality.

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
