# Peer review of "Reframing Communication about Fall Prevention Programs to Increase Older Adults’ Intentions to Participate"

_ijerph, 2024, doi:10.3390/ijerph21060704_

Round 1
Reviewer 1 Report
Comments and Suggestions for Authors
See attached Review Report Word file

See attached Review Report Word file
Reviewer 2 Report
Comments and Suggestions for Authors
This paper investigates a simple research question.
The strenght of this paper is the method in which precisely and systematic the data is collected and analysed. Especially I liked the qualitative inductive thematic analysis apart from the several statistical analysis. So the collected data are well used. The weakness lies in the fact the authors allready mention in the limitation paragraph": probably too much high educated people inside the sample and no attempt was tried to get people from the lower regions in terms of education and income and/or zip-codes.
About the sample I agree with the limitations mentioned : the sample may have a somewhat lower representation regarding the Dutch 65+ people inclusive SES-factors: low literated people might be excluded because of the language issue that was investigated and because of the reading and writing requirements.
I would have liked to see more effort to get low literated people and people with a lower SES rate inside the sample.
Reviewer 3 Report
Comments and Suggestions for Authors
Dear Authors,
Please, see my comments below.
- The introduction is satisfatory.
- Please, insert more about the larger project, as a link for more information.
- The sample size calculation lacks information. To an "a priori" estimation, the effect size is needed (but it was not reported). Was it a 2-tailed sample size estimation? What online calculation tool was used? Please, clarify.
- The statistical section has some issues to be addressed. 1st: how did you manage the multiple testing issue as you performed several tests to be related to each other? Did you consider another testing procedure? Did you consider any log transformation or z-score to allow the repeated measures comparison?
- The actual sample size was smaller than the calculated. Even calculating the post hoc power, the procedure you established was not achieved. How you expected to generalize the results if the basis was not followed accordingly?
- The discussion section would benefit of another organization. Summarize the results first, then compare them to the available literature. Also, include your own explanation about the findings. Explain how the findings are novel and enhances the practical management of falls in older people. Then, set the limmitations (as you already did), and conclude.
- Raw data should be available through an open repository.
- The supplemental material was not declared accordingly. Please, review the final statement for that.
Round 2
Reviewer 3 Report
Comments and Suggestions for Authors
Dear authors
I have read the report you provided, but I could not see the differences marked across the text.
You must hightlight all changes so the reviewer is able to reassess the draft.
Regards.